# Arthroscopy-Assisted Reduction and Internal Fixation versus Open Reduction and Internal Fixation for Glenoid Fracture with Scapular Involvement: A Retrospective Cohort Study

**DOI:** 10.3390/jcm11041131

**Published:** 2022-02-21

**Authors:** I-Hao Lin, Tsung-Li Lin, Hao-Wei Chang, Chia-Yu Lin, Chun-Hao Tsai, Chien-Sheng Lo, Hui-Yi Chen, Yi-Wen Chen, Chin-Jung Hsu

**Affiliations:** 1Department of Orthopedics, China Medical University Hospital, Taichung 40447, Taiwan; d24686@mail.cmuh.org.tw (I.-H.L.); d18144@mail.cmuh.org.tw (T.-L.L.); d22067@mail.cmuh.org.tw (H.-W.C.); d29585@mail.cmuh.org.tw (C.-Y.L.); d7940@mail.cmuh.org.tw (C.-H.T.); 2Department of Sports Medicine, College of Health Care, China Medical University, Taichung 406040, Taiwan; 3Graduate Institute of Biomedical Sciences, China Medical University, Taichung 40402, Taiwan; evinchen@mail.cmu.edu.tw; 4Department of Orthopedics, Chung Shan Medical University Hospital, Taichung 40201, Taiwan; johnlcs317@gmail.com; 5Department of Medical Image, China Medical University Hospital, Taichung 40447, Taiwan; d7396@mail.cmuh.org.tw; 6X-Dimension Center for Medical Research and Translation, China Medical University Hospital, Taichung 40447, Taiwan; 7Department of Bioinformatics and Medical Engineering, Asia University, Taichung 41354, Taiwan; 8School of Chinese Medicine, China Medical University, Taichung 404333, Taiwan

**Keywords:** arthroscopy, Ideberg, scapula fracture, glenoid fracture, shoulder surgery

## Abstract

Background: We investigated the superiority of arthroscopy-assisted reduction and internal fixation (ARIF) to open reduction and internal fixation (ORIF) for treating glenoid fracture with scapular involvement. Methods: We retrospectively enrolled patients with glenoid fracture who underwent ARIF or ORIF from 2010–2020. Radiographic outcomes were assessed, and clinical outcomes (active range of motion [ROM], visual analog scale [VAS], Constant, and Disabilities of the Arm, Shoulder and Hand [DASH]) were evaluated 12 months postoperatively. Results: Forty-four patients with Ideberg type II–VI glenoid fractures (ARIF: 20; ORIF: 24; follow-up 12–22 months) were included. Union was achieved in all patients. Active ROM values were comparable between the approaches. Constant and DASH scores were non-significantly better with ARIF (90.9 ± 9.2 vs. 86.6 ± 18.1 [*p* = 0.341] and 6.8 ± 9.4 vs. 9.3 ± 21.3 [*p* = 0.626], respectively). However, VAS scores were significantly lower with ARIF (1.5 ± 0.6 vs. 2.7 ± 1.4, *p* = 0.001). Associated intra-articular lesions (articular depressions [80%], superior labral anterior-posterior tear [20%], labral tears [30%]) were found in most ARIF cases and were repaired during ARIF. Conclusions: For glenoid fracture with scapular involvement, ARIF allows accurate diagnosis of fracture pattern and the management of associated intra-articular lesions, with better pain control outcomes than ORIF. Thus, arthroscopy-assistant surgery should be considered in patient with glenoid fracture.

## 1. Introduction

Glenoid fracture with scapular involvement is a rare but challenging clinical problem [1]. Scapular fractures comprise 0.4 to 1 percent of all fractures, and 10 percent of scapular fractures involve the glenoid [2,3,4]. Ideberg et al. classified glenoid fractures into six types based on plain radiology films: type I glenoid fracture is glenoid fracture without scapular involvement (anterior or posterior rim bony Bankart lesion) and types II to VI glenoid fractures are fractures with scapular involvement, with increasing complexity from type II to VI [2]. There are controversies regarding the treatment of glenoid fractures, which depend on the degree of displacement, the fracture gap, and the fragment size [5]. When the articular fragment constitutes more than 20% of the glenoid cavity, surgical intervention for reduction and stabilization may be considered to prevent the development of glenohumeral instability and osteoarthritis over time [6]. Moreover, most glenoid fractures were the result of high-energy injuries and were often associated with other injuries, including Hill-Sachs lesions and clavicular or acromial fracture, which require surgical management [6,7].

Open reduction and internal fixation (ORIF) has been the standard treatment for intra-articular fracture of the glenoid [7]. Judet approach allows access to the entire posterior aspect of the scapular body but requires a large skin incision and extensive muscular disruption [8]. Gauger et al. described a modified Judet approach, which involves minimized soft tissue trauma; however, the technique requires significant soft tissue dissection for articular exposure, with the risk of injury to neurovascular structures (particularly the suprascapular nerve), potentially resulting in postoperative weakness and stiffness [9]. Furthermore, direct visualization of the glenoid articular surface remains challenging in these traditional open surgical approaches [10].

To overcome these shortcomings of open surgery, Cameron et al. described the use of arthroscopy-assisted reduction and internal fixation (ARIF) for Ideberg type I fractures [11]. There are also few case reports of the ARIF technique for Ideberg type III and V fractures in the literature [12,13,14]. Although Bonnevialle et al. reported the outcome of ARIF, with considerable advances over ORIF and the reduction of complications and reoperation rates, the study focused on Ideberg type IA (anterior glenoid rim fractures) alone [15]. To the best of our knowledge, no prior study has directly compared ARIF and ORIF in glenoid fractures other than Ideberg type I. Moreover, Ideberg type II to VI fractures, which are glenoid fractures with scapular involvement, are more complicated than type I fractures. Thus, a study that compares the results of ARIF and ORIF for glenoid fracture with scapular involvement seems critical.

The aim of this study was to compare the radiographic and clinical results of ARIF and ORIF for the treatment of glenoid fracture with scapular involvement (Ideberg type II to VI), and to report findings of the intra-articular lesion and the management of the ARIF group. We hypothesized that ARIF may had better outcome than ORIF in glenoid fracture with scapular involvement.

## 2. Materials and Methods

### 2.1. Patients

This study was approved by the local institutional review board (CMUH102-REC2-062), and all study participants provided informed consent. The study population was derived from a retrospective database that included adult patients who were surgically treated for glenoid fracture between January 2010 and December 2020 and received a minimum follow-up of 1 year. The inclusion criteria were glenoid fracture with >20% articular involvement, 5 mm articular displacement, and scapular involvement. The exclusion criteria were: (1) glenoid fracture without scapular involvement (Ideberg type I); (2) open glenoid or scapular fractures; (3) neurologic deficit due to major nerve injury of the ipsilateral limb; and (4) less than 1 year of follow-up.

Included patients were categorized into two groups based on the surgical technique—ORIF or ARIF. Conventional ORIF was the first cohort (January 2010 and December 2015), and ARIF was the second cohort (January 2015 and December 2020). This is because of the study design that all surgeries were performed by the same four surgeons over this period.

### 2.2. Preoperative Evaluation

All patients underwent radiographic examination (true AP, Y scapular view) and computed tomography (CT) with three-dimensional (3D) reconstruction of the shoulder before the operation. The Ideberg classification of glenoid fractures was assessed by a musculoskeletal radiologist (H.-Y.C.), and all radiographic data were recorded [2].

### 2.3. Surgical Technique

Surgery was performed by four consultants (C.-J.H., C.-H.T., C.-S.L., and T.-L.L.) specialized in upper limb surgery in all cases, under general anesthesia. All patients in the ORIF and ARIF groups underwent the same treatment protocol of open surgery for associated injuries, such as scapular body fracture, clavicular fracture, and acromioclavicular joint injury.

In the ORIF group, patients were placed in the decubitus position for Judet approach. Through deep dissection and elevation of the posterior deltoid off the scapular spine, with superior dissection of the infraspinatus inferiorly or posterior splitting of the infraspinatus and teres minor interval, the joint was exposed for open reduction, followed by osteosynthesis.

In the ARIF group, arthroscopy-assisted surgery for the glenoid part followed that of the ORIF. Patients were placed in two different positions, depending on the surgery. If fixation of the scapular body was required (Ideberg scapular type IV and V), the patient was placed in the decubitus position for Judet approach. After the completion of osteosynthesis, further arthroscopy was performed in the same decubitus position; however, if fixation was not required for the scapular fracture (Ideberg scapular types II, III, and VI), the patient was placed in the beach chair position. After fixation for accompanying fracture and injury (osteosynthesis of acromioclavicular joint injuries or clavicular fractures), arthroscopy was performed.

For the arthroscopy, a traction device (SPIDER2 Limb Positioner, Smith&Nephew, London, UK) was used. The posterior portal was first made to serve as the viewing portal. Initially, sufficient irrigation using normal saline was performed to remove intra-articular hematoma for further fracture evaluation. To identify the fracture line, hematoma, broken cartilage, and fragments were debrided using an arthroscopic shaver. Reduction of the fragment was achieved with a probe and was maintained with 1.5 Kirschner wires percutaneously. For definite fragment fixation, cannulated or headless screws (Acutrak 2 Headless Compression Screw, Acumed, Hillsboro, OR, USA) were used. If concomitant lesions of the periarticular soft tissue, including superior labral anterior-posterior (SLAP) tears and labral tears were observed, these intra-articular lesions were repaired simultaneously, using a suture anchor. The posterior, lateral subacromial, and anterior (rotator interval and biceps accessory) portals were sufficient for all of the arthroscopy procedure. No Neviaser portal was created in our cases.

### 2.4. Postoperative Protocol

All shoulders were immobilized with a sling postoperatively for 4 weeks. At 4 weeks after surgery, passive range of motion (ROM) had gradually increased to 90° of forward elevation, 90° of abduction, and 10° of external rotation. After 6 weeks, active ROM exercises in all directions and partial strength exercises were permitted in most patients.

### 2.5. Evaluation

The medical records and radiographs of all patients were reviewed. Outpatient follow-up was conducted at 3 weeks, 6 weeks, 3 months, 6 months, and 12 months after surgery. Union was defined as the detection of callus formation on anterior-posterior and lateral radiographs [16], and this was accessed by the same musculoskeletal radiologist. Active ROM (forward elevation, lateral elevation, external rotation, and internal rotation), visual analog scale (VAS) score, Constant-Murley Shoulder Outcome (Constant) score, and Disabilities of the Arm, Shoulder, and Hand (DASH) score of the injured limb were recorded at 12 months postoperatively.

### 2.6. Statistical Analysis

Statistical analysis was performed using IBM SPSS Statistics (version 23.0; SPSS Inc., Armonk, NY, USA). Data are presented as averages and standard deviations. Descriptive statistics are presented as means and 95% confidence intervals (CIs) for continuous variables and as counts and percentages for categorical variables. The independent samples t-test was used to compare active ROM, VAS, Constant score, and DASH score, with *p* < 0.05 considered to indicate statistical significance.

## 3. Results

### 3.1. Data Presentation and Population Comparison

Forty-four patients (34 men, 10 women) were enrolled. There were 20 and 24 ARIF and ORIF patients, respectively. The mechanism of injury was high-energy trauma in all 44 patients (38 motor vehicle accidents and 6 cases of fall from a height). The Strengthening the Reporting of Observational Studies in Epidemiology (STROBE) flowchart detailing the study design is shown in Figure 1. There were no differences in demographic data, Ideberg classification, combined injury, and concomitant injury other than shoulder between patients in the ARIF and ORIF groups (Table 1). The mean follow-up time was 15 months (range, 12–22 months). Union was achieved in all patients, no patient was lost to follow-up, and there was no fixation failure or neurovascular complications.

### 3.2. Outcome Comparison

Active ROM was comparable between the groups (Table 2).

Mean VAS score was significantly lower in the ARIF group than in the ORIF group. Constant and DASH scores were better in the ARIF group than in the ORIF group, although the difference was not statistically significant (Table 3).

### 3.3. Arthroscopic Finding in ARIF and Case Presentation

In the ARIF group, arthroscopy showed articular surface depression in 16 cases (80%) (Figure 2). Six patients (30%) had labral tears, three of whom underwent labral repair using suture anchors, while the two other patients underwent chondral shaving for the frayed edge (Figure 3). Four patients (20%) had superior labrum anterior to posterior (SLAP) lesions and underwent primary repair using suture anchors (Figure 4). All arthroscopic findings and procedures are summarized in Table 4.

## 4. Discussion

This is the first retrospective study to compare ARIF and ORIF in patients who have glenoid fracture with scapular involvement. Our results demonstrate comparable active ROM, Constant, and DASH scores between groups. However, less postoperative pain was observed in the ARIF group, with a significantly lower VAS score, than in the ORIF group. Additionally, associated intra-articular lesions were found in a majority of ARIF cases, and primary repair was performed simultaneously.

The choice of open surgical approach for glenoid fracture, including the anterior deltopectoral approach [16,17,18], superior approach through the rotator cuff interval or acromial approach [19], or Judet approach, is influenced by the fracture pattern. However, extensive soft tissue dissection, impaired blood supply to the fragments, and postoperative muscle weakness, scar tissue adhesion, or stiffness remain major concerns with these open approaches [20,21,22,23]. Regarding glenoid fragment assessment in the open approach, Rongguang et al. indicated that only less than 50% of the glenoid rim could be exposed with the single deltopectoral or Judet approach in a cadaveric study. Particularly, the superior part of the glenoid fracture is difficult to approach via these approaches [24].

Scheibel et al. reported excellent clinical outcomes of ARIF for anterior glenoid rim fractures [25]. However, there have been few case reports or comparative studies on different fracture patterns and treatments. Bonnevialle et al. compared ARIF and ORIF for anterior glenoid rim fractures and reported similar functional outcomes between both procedures, but fewer complication and reoperation rates for ARIF [15]. However, these studies only focused on anterior rim glenoid fracture (Ideberg type I). To the best of our knowledge, no prior studies have directly compared ARIF to ORIF in glenoid fracture other than Ideberg type I.

In our study, favorable functional outcomes were observed in the ARIF group. However, there was no significant difference between the ARIF and ORIF groups, which may be due to the great healing potential of glenoid fractures, although complete reduction may not have been achieved in the ORIF group. Several studies on ORIF for glenoid fracture concluded that open surgical treatment yields good-to-excellent results if there are no postoperative complications or permanent brachial plexus injury [26,27,28]. Furthermore, Mayo et al. documented that poor outcome mainly resulted from associated nerve injuries or poor rehabilitation, rather than from glenohumeral arthritis [29]. Thus, incomplete articular reduction is not considered a major problem. Moreover, functional outcome does not seem to be related to fragment reduction as because the glenohumeral joint is regarded as a non-weight bearing joint.

In our series, compared to the ORIF group, better outcome in terms of shoulder pain, which was measured with the VAS scale, was observed in the ARIF group. This may be due to the less soft tissue dissection in the ARIF group compared to that in the ORIF group. In the ARIF group, although Judet approach was also performed for osteosynthesis of the scapular body, soft tissue exposure around the glenoid joint, including the joint capsule and posterior glenohumeral ligament, could be avoided. In addition, the violation of the neurovascular bundle, which loops through the spinoglenoid notch, could be avoided.

Furthermore, with arthroscopy, intra-articular lesions can be inspected directly. Primary repair for intra-articular soft tissue injuries, such as SLAP lesions and labral tear, could be performed at the same time. There were 80% glenoid articular surface depressions, 30% labral tears, and 20% SLAP tears in ARIF group, and all cases could be treated simultaneously with arthroscopic reduction and fixation techniques. Reduction of articular step-off could be achieved, which may have prevented further traumatic arthritis [11]. Primary repair of the intra-articular soft tissue injuries could have also contributed to the better VAS outcomes in our ARIF group.

This study had some limitations. First, our sample size was small because of the rarity of glenoid fractures. Determining which procedure is superior would require a higher powered prospective randomized control study with large sample size. Second, the complexity of the fracture pattern varied from Ideberg type II to type VI; although there was no significant difference between the ARIF and ORIF groups regarding fracture pattern, the complexity of the fracture pattern and the additional procedure may have affected the outcomes. Third, regarding the better finding of VAS at post operative 12 months, though it was statistically significantly different, the result of 1.5 versus 2.75 may not be clinically different. Fourth, longer follow-up would be required.

## 5. Conclusions

For glenoid fracture with scapular involvement, ARIF allows accurate diagnosis of fracture pattern and the management of associated intra-articular lesions, with better pain control outcomes than ORIF. Thus, arthroscopy assistant surgery should be considered in patient with glenoid fracture.

## Figures and Tables

**Figure 1 jcm-11-01131-f001:**
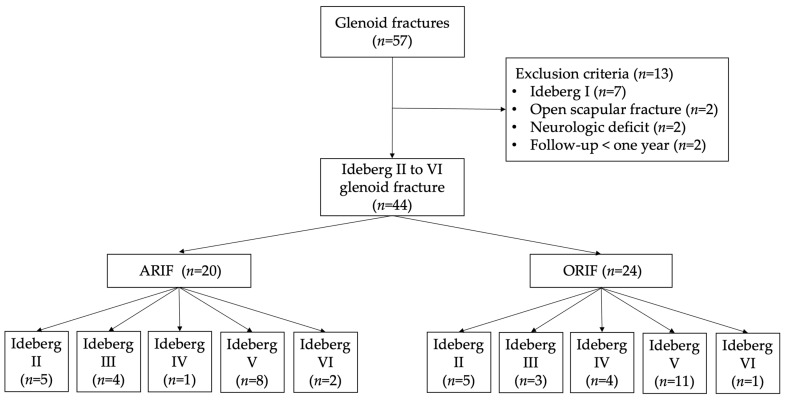
STROBE flowchart detailing the design of the study.

**Figure 2 jcm-11-01131-f002:**
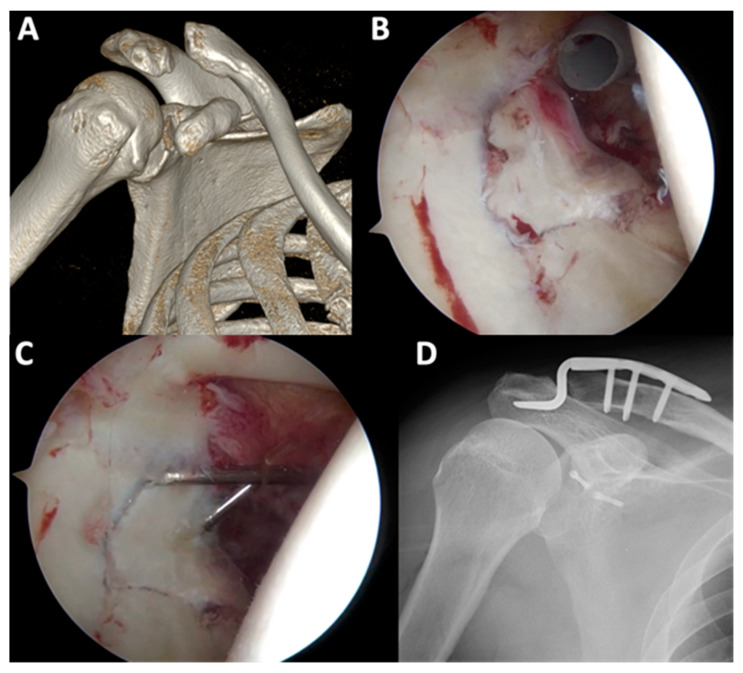
In patient no. 7 with Ideberg classification type III (**A**), after the hematoma was debrided and chondral shaving was performed, fragment reduction was achieved using the probe (**B**). Fragment fixation was performed with transcutaneous screwing using 2.7 mm cannulated screws (**C**,**D**).

**Figure 3 jcm-11-01131-f003:**
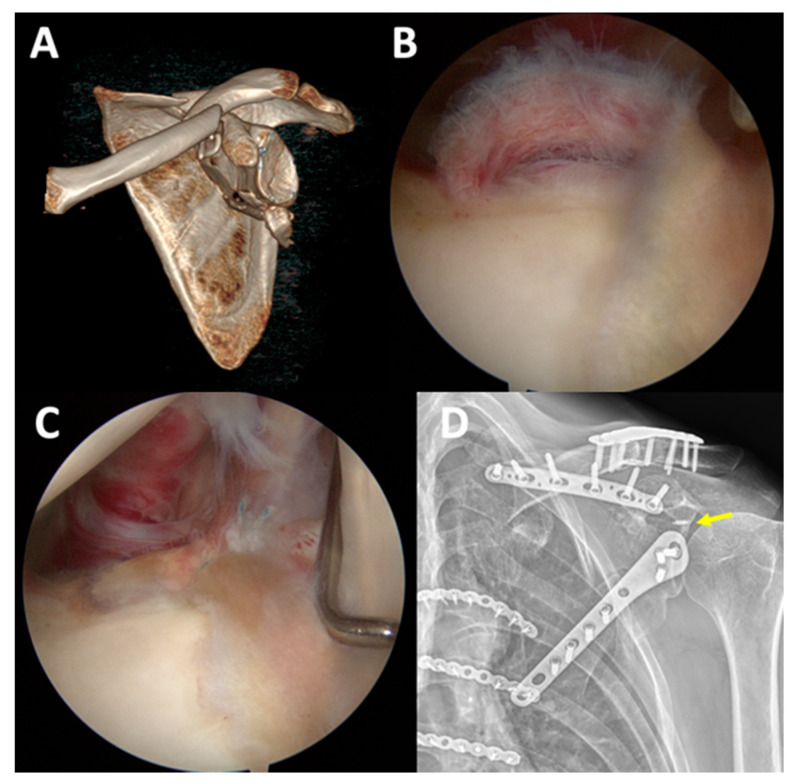
In patient no. 6 with Ideberg type II fracture (**A**), after osteosynthesis for scapular body fracture, arthroscopy showed a posterior-superior labral tear (**B**). Primary repair of the tear with a suture anchor was performed ((**C**,**D**), yellow arrow indicates the suture anchor in a postoperative radiograph).

**Figure 4 jcm-11-01131-f004:**
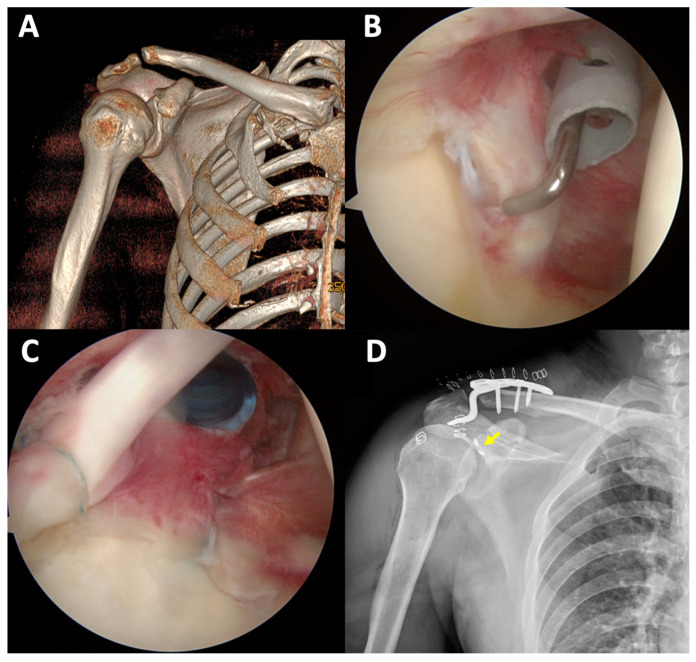
In patient no. 9 with Ideberg type III fracture (**A**), a SLAP lesion was observed (**B**). Primary repair with SLAP repair was performed using suture anchors ((**C**,**D**), yellow arrow indicates a suture anchor in a postoperative radiograph).

**Table 1 jcm-11-01131-t001:** Demographic data of patients with ARIF and ORIF.

	Surgery Type		
Variables	ARIF (*n* = 20)	ORIF (*n* = 24)	*p* Value
Age, years (95% CI)	41.15 (25–61)	43.12 (21–65)	0.667
Gender in female, *n* (%)	15 (75.0)	19 (79.2)	0.743
Motor vehicle accidents, *n* (%)	16 (80.0)	22 (91.7)	0.129
Right laterality, *n* (%)	8 (40.0)	10 (41.7)	0.563
Dominant hand in right, *n* (%)	20 (100.0)	23 (95.8)	0.890
Ideberg classification			
II, *n* (%)	5 (25.0)	5 (20.8)	0.617
III, *n* (%)	4 (20.0)	3 (12.5)	0.774
IV, *n* (%)	1 (5.0)	4 (16.7)	0.128
V, *n* (%)	8 (40.0)	11 (45.8)	0.694
VI, *n* (%)	2 (10.0)	1 (4.2)	0.826
Combined injury			
Clavicle fracture, *n* (%)	7 (35.0)	7 (29.2)	0.633
Acromion fracture, *n* (%)	2 (10.0)	2 (8.3)	0.704
Coracoid fracture, *n* (%)	1 (5.0)	1 (4.2)	0.896
Acromioclavicular injury, *n* (%)	4 (20.0)	3 (12.5)	0.741
Nil, *n* (%)	6 (30.0)	11 (45.8)	0.287
Concomitant injury other than shoulder			
Head injury, *n* (%)	2 (10.0)	2 (8.3)	0.712
Hemopneumothorax, *n* (%)	4 (20.0)	3 (12.5)	0.807
Upper extremity fracture, *n* (%)	3 (15.0)	2 (8.3)	0.760
Pelvic fracture, *n* (%)	1 (5.0)	1 (4.2)	0.729
Lower extremity fracture, *n* (%)	2 (10.0)	2 (8.3)	0.761
Nil, *n* (%)	8 (40.0)	14 (58.4)	0.288

ARIF, arthroscopy-assisted reduction and internal fixation; ORIF, open reduction and internal fixation; CI: confidence interval.

**Table 2 jcm-11-01131-t002:** Active range of motion comparison.

Range of Motion	ARIF	ORIF	*p* Value
Forward elevation, ° (mean)	175.5 ± 10.9	166.25 ± 33.0	0.239
Lateral elevation, ° (mean)	168 ± 20.4	165 ± 31.8	0.719
External rotation, ° (mean)	83.25 ± 7.6	80.41 ± 16.7	0.489
Internal rotation (spine level)	T_10±2_	T_11±3_	0.099

**Table 3 jcm-11-01131-t003:** Functional outcome comparison.

	ARIF	ORIF	p Value
VAS	1.5 ± 0.6	2.75 ± 1.45	0.001
Constant score	90.95 ± 9.2	86.62 ± 18.2	0.341
DASH score	6.84 ± 9.5	9.37 ± 21.3	0.626

**Table 4 jcm-11-01131-t004:** Details of ARIF group.

Patient No.	Arthroscopic Finding	Arthroscopic Procedure
1	Articular surface depressed	ARIF with 3.0 mm cannulated screw
2	Articular surface depressed	ARIF with 3.0 mm cannulated screw
3	Articular surface depressed	Chondral shaving
4	Articular surface depressed	ARIF with 3.0 mm cannulated screw
5	Articular surface depressed	ARIF with 2.4 mm headless screws
6	Articular surface depressed	ARIF with 3.0 mm cannulated screw
7	Articular surface depressed	ARIF with 3.0 mm cannulated screw
8	Articular surface depressed	ARIF with 3.0 mm cannulated screw
9	Articular surface depressed	ARIF with 3.0 mm cannulated screw
10	Articular surface depressed	Chondral shaving
11	Articular surface depressed, labrum tear	Chondral shaving
12	Articular surface depressed, labrum tear	Labrum repair with 2.8 mm suture anchor
13	Articular surface depressed, labrum tear	Labrum repair with 2.8 mm suture anchor
14	Articular surface depressed, labrum tear	ARIF with 3.0 mm cannulated screw
15	Articular surface depressed, labrum tear	ARIF with headless screws, labrum repair with 2.8 suture anchor
16	Articular surface depressed, labrum tear	labrum repair with 2.8 suture anchor
17	SLAP lesion	SLAP repair with Y-Knot^®^ RC anchors
18	SLAP lesion	SLAP repair with Y-Knot^®^ RC anchors
19	SLAP lesion	SLAP repair with Smith & Nephew TWINFIX anchors
20	SLAP lesion	SLAP repair with Y-Knot^®^ RC anchors

ARIF, arthroscopy-assisted reduction and internal fixation; SLAP, superior labrum anterior to posterior.

## Data Availability

All the available data have been presented in this study. Details, regarding where data supporting reported results can be requested at the following e-mail address: jeffrey59835983@gmail.com.

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
