# Peer review of "Arthroscopy-Assisted Reduction and Internal Fixation versus Open Reduction and Internal Fixation for Glenoid Fracture with Scapular Involvement: A Retrospective Cohort Study"

_jcm, 2022, doi:10.3390/jcm11041131_

Round 1

Reviewer 1 Report

The article may be of interest, since it compares two different techniques, although it must be taken into account that the sample is small in order to obtain strong conclusions.

table 1, I doubt that it is necessary, in any case, if you decide to introduce, it is necessary to group or make a synthesized table, a table of that size is not relevant.
On the other hand, the conclusions must be written in a way that suggests, not affirms.

Reviewer 2 Report

General comments:

The authors aim to to compare the radiographic and clinical results of ARIF and ORIF for the treatment of glenoid fracture with scapular involvement and to report findings of the intra-articular lesion and the management of the ARIF group.

First the authors must be congratulated for having collected such an interesting series.

The manuscript could be improved and notably:

  • There is information that are not positioned in the good section (population description)
  • Many redundance between the text and the tables but no analysis of the associated intra articular lesions.
  • Results and Discussion must be reorganized in a 2 or 3 parts sections to make the article more readable
  • Discussion is too long especially in its first part and must be reconsidered and rewritten to highlight the most interesting message. Too many information are not related to the article and must be removed or shortened.

Specific comments:

Introduction

What was the hypothesis ?

Methods

Line 35 : remove the decimales for VAS, Constant and DASH scores

Results

As it is a retrospective study, the description of the population and the two groups must appear in the Methods section, not in the Results. In the same way, the Tables 1 and 2 must be position in the Methods section.

Line 190 to 204: no redundant information must appear between the text and the Tables.

Line 519-520: “Constant and DASH scores were better in the ARIF group than in 519 the ORIF group, although the difference was not statistically significant.” To be removed: not significant than no difference.

Line 521-523: No interest in giving an analysis of a non-significant difference, the readers can see them in the Tables.

Line 531: why to provide specific information on this specific case ?

A specific focus must be done on the treatment of the associated lesions

Table II: the follow-up must be given for each group and compared between them.

Discussion

The discussion must begin by the answer to the hypothesis and the main take-home messages of the article.

The discussion must be organized and structured with 2 to 3 sections, for example “pain”, “function”, “management of intra-articular lesion”

Then limits and conclusion.

The first part of the conclusion is too long and make the history of the scapular fixation and the limits of the approaches. However, the article must keep focused on its main topic and comparison between the two methods and must be organized to the light of the found results.

Round 2

Reviewer 1 Report

The authors have made the requested changes.

Recommend these references:

10.3390/medicine58020227

10.3390/jcm10112315

Author Response

This manuscript is a resubmission of an earlier submission. The following is a list of the peer review reports and author responses from that submission.

Round 1

Reviewer 1 Report

reduction and internal fixation for glenoid fracture with scapular involvement: a retrospective cohort study

The aim of this study was to compare the radiographic and clinical results of ARIF and 80 ORIF for the treatment of glenoid fracture with scapular involvement (Ideberg type II to VI), and to report findings of the intra-articular lesion and the management of the ARIF group. To the best of your knowledge, no prior study has directly compared ARIF and ORIF in glenoid fractures other than Ideberg type I.

Line 104: please add the citation of the Ideberg classification.

Line 108: how do you select patients for each procedure?

Retrospective studies have confounding by indication.

 Line 146; if possible, please add the operative time and blood loss on each procedure.

Line 238: VAS at 12 months, 1.4 versus 2.4.

         I think it is statistically significantly different, but it is not clinically different. Therefore, please weaken your message on VAS.

Line 319: I agree with your limitation. Very small sample size did not reveal any the results of the comparison.

Line 330: Diagnosis of fracture reduction is not outcome here, please split your results in your study and your interpretation. Therefore, add the results to your outcome.

Reviewer 2 Report

It's a very interesting study, but I think there are many problems.

Small sample size.

Poor novelty.

lack of main findings.

Reviewer 3 Report

General Comments

this is a retrospective study evaluating the results of surgical treatment in a series of patients with glenoid fractures.

The major problems are the retrospective nature of the study and the complete lack of randomization.

The advantage of the study is the high number of patients included, i.e. 36 and the sufficient follow up.

Abstract

it is well organized and focused.

Introduction

Sufficient length and structure.

Results

Clear presentation.

Tables and Figures

Sufficient quality and presentation.

Discussion

The limitation paragraph is well documented.

Round 2

Reviewer 1 Report

As the other reviewers pointed out, very small sample size did not reveal any the results of the comparison.

Therefore, the results is less clinical implication, and do not change the clinical practice.

Reviewer 2 Report

Thank you for the correction.

I think it has been sufficiently revised and is suitable for acceptance.